# Distributed Network-Constrained P2P Community-Based Market for Distribution Networks

Carlos Oliveira [1,2], Micael Simões [1,2], Leonardo Bitencourt [3], Tiago Soares [1,*] and Manuel A. Matos [1,2]

1 Center for Power and Energy Systems, Institute for Systems and Computer Engineering, Technology and Science (INESC TEC), 4200-465 Porto, Portugal
2 Faculty of Engineering, University of Porto, 4200-465 Porto, Portugal
3 Electrical Energy Department, Federal University of Juiz de Fora (UFJF), Juiz de Fora 36036-330, MG, Brazil
* Correspondence: tiago.a.soares@inesctec.pt

**Abstract:** Energy communities have been designed to empower consumers while maximizing the self-consumption of local renewable energy sources (RESs). Their presence in distribution systems can result in strong modifications in the operation and management of such systems, moving from a centralized operation to a distributed one. In this scope, this work proposes a distributed community-based local energy market that aims at minimizing the costs of each community member, accounting for the technical network constraints. The alternating direction method of multipliers (ADMM) is adopted to distribute the market, and preserve, as much as possible, the privacy of the prosumers' assets, production, and demand. The proposed method is tested on a 10-bus medium voltage radial distribution network, in which each node contains a large prosumer, and the relaxed branch flow model is adopted to model the optimization problem. The market framework is proposed and modeled in a centralized and distributed fashion. Market clearing on a day-ahead basis is carried out taking into account actual energy exchanges, as generation from renewable sources is uncertain. The comparison between the centralized and distributed ADMM approach shows an 0.098% error for the nodes' voltages. The integrated OPF in the community-based market is a computational burden that increases the resolution of the market dispatch problem by about eight times the computation time, from 200.7 s (without OPF) to 1670.2 s. An important conclusion is that the proposed market structure guarantees that P2P exchanges avoid the violation of the network constraints, and ensures that community agents' can still benefit from the community-based architecture advantages.

**Keywords:** energy communities; distributed energy resources; energy trading; distributed optimization; optimal power flow

## 1. Introduction

### 1.1. Background and Motivation

The increasing penetration rate of distributed energy resources (DERs) and the rapid advance of technology, such as residential photovoltaics (PV) panels and energy storage systems (ESS), led to a rise in prosumers' numbers and unlocked new market opportunities for diverse investors in the distribution network [1,2]. This leads to the design of new market platforms that can empower end-users to actively participate with other market participants by sharing and using clean and local energy [2].

Local energy communities (LECs) are a new concept that has been introduced to the electric distribution grid in the last decade [3]. LECs emerged from the need for new consumer-centric market designs (e.g., peer-to-peer market (P2P)) focused on the welfare of the community [4]. Community participants can decide to share access to a shared resource and strive toward a common goal [4]. Some studies suggest that P2P trading improves self-consumption, promotes the use of local DERs, and reduces electricity costs [5–7].

Despite the benefits DERs can bring to prosumers and community welfare, their erratic nature can create an energy surplus. In conventional grids, prosumers can trade the excess energy, store the energy, or sell it to the grid [8,9]. Setting a P2P market and defending secure, trustworthy, and optimal function of the complex energy system is challenging. If LECs take place within a distribution network, the following question can be raised: "How can this market structure be integrated in the current systems without violating operational or technical constraints? Will it induce more grid losses, grid congestion, or other physical constraints to DSO operations?" Hence, this paper proposes a mathematical model to reduce the operational costs of a community. The problem is decomposed into prosumers and solved using the alternating direction method of multipliers (ADMM), while preserving the privacy of each agent. The mentioned model integrates network constraints and the relaxed branch flow model is adopted.

## 1.2. Literature Review

The operation of an energy community needs the introduction of an energy management system (EMS) to optimize the utilization of open resources. The EMS scheduling function is either structured as a (i) centralized, (ii) partially distributed (so-called decentralized) or (iii) distributed optimization framework [3].

The centralized framework (i), consists of a single central node called the community manager that coordinates members' assets, provides services to the DSO, and communicates with different existing markets linked to the LEC, as in [10–12]. In (ii), unlike the centralized approach, each participant in the LEC is independent and has its own EMS in a decentralized optimization approach. The LEC problem can be decomposed into N subproblems. N corresponds to the number of LEC members, and each member can solve its own problem independently. Methods to solve decomposable optimization problems use derivate-based approaches, such as, distributed gradient descent (DGD) [13], primal Benders decomposition approach [14], or ADMM. ADMM is the most frequently adopted consensus algorithm [15–18]. In this framework, a supervisor node constrains the power balance of each node that acts in a distributed manner [3]. Lastly, (iii) entails that the LEC optimization problem is solved individually without any central node to check the power balance [3,19], thus every peer communicates directly to its peers.

Lilla et al. [20] proposed a model that aims to minimize the LEC costs assuming power losses in the community's network. These losses are attributed to the energy exchange from one prosumer to another or between a prosumer and the supply network. The distributed procedure is divided into two phases. In the first phase, the estimation of the power losses is made for each prosumer, while in the second phase network power losses are recalculated for each power transaction. Similar works [21,22] additionally consider local generation and energy storage systems, and the option to trade with an external supplier. However, line losses are not taken into account in the optimization procedure.

Silvestre D. et al. [23] presents the application of blockchain technology for energy exchange and evaluates the power losses caused by the exchange. However, grid operational constraints are not considered. More recently, Münsing et al. [24] used ADMM to solve an optimal power flow (OPF) with the branch flow model in a distributed fashion for a microgrid network on a blockchain platform. This work is based on OPF for microgrids where the community concept is not considered. The authors in [25] implement a two-level optimization problem for a collaborative microgrid, where the lower level first optimizes the dispatch of assets and performs market clearing. At the upper level, the problem then takes the standpoint of the collaborative microgrid operator. In addition, Ref. [26] formulated a model as a stochastic bi-level problem. At the upper level, the DSO guarantees that the network constraints, such as power flows and voltage magnitudes, are respected. At the lower level, the microgrids minimize the costs of the individual nodes.

Guerrero et al. [2], extended the decentralized P2P energy trading system considering network constraints at the distribution level. Trades are validated during the bidding process based on the network constraints. Li et al. [27] applied Nash bargaining theory to

the bilateral energy trading (BET) problem with the distribution system OPF and solved the decomposed BET problem with ADMM.

Bilateral and full P2P exchanges are quite similar, except when agreements are made in bilateral exchanges. In both markets, there is complete freedom of choice and autonomy, with decision-making power resting with consumers. However, it is challenging to integrate these systems into existing infrastructure, as they require high levels of investment and maintenance. As for bilateral trading arrangements, they can often trigger competing bilateral arrangements between other market participants. In the case of community-based markets, this can be said to: (*i*) improve relationships and engagement among community members as they share a common asset; and (*ii*) provide potential new services to network operators provided by the community manager. However, disadvantages include the need to achieve the energy consumption preferences of all community members at all times, the need to aggregate data from all members, and the need to manage members' expectations of the community manager [28].

Table 1 presents the analyzed studies that focus on the impact of P2P trading on the network. There are studies that include network constraints in the market clearing model, while others use sensitivity coefficients, such as voltage sensitivity or power transfer distribution factor, in order to estimate potential problems in the network. There are also studies that separate the dispatch of the market from the assessment of the impact of P2P exchanges on the network.

**Table 1.** Literature review on related works.

| Refs. | Market Structure | Power Flow Type | Scheduling Structure | Optimization Type |
|---|---|---|---|---|
| [1] | Community | No OPF | Distributed | Bi-level |
| [6] | Community | AC-OPF | Centralized | Bi-level |
| [20] | Community | Net losses | Distributed | Bi-level |
| [29] | Fully P2P | AC-OPF | Centralized | Integrated |
| [30] | Fully P2P | AC-OPF | Centralized | Bi-level |
| [2] | Fully P2P | AC-OPF | Distributed | Integrated |
| [26] | Microgrid | DistFlow | Distributed | Bi-level |
| [31] | Microgrid | AC-OPF | Distributed | Bi-level |
| [4,24] | Micro grid | AC-OPF | Distributed | Integrated |
| [27] | BET | AC-OPF | Distributed | Integrated |
| Proposed work | Community | AC-OPF | Distributed | Integrated |

One important part of performing a fully distributed P2P market is related to the financial transactions system. Due to the high level of transactions in this type of market, the system may consider adapting current financial systems or the use of cryptocurrency. As this is not the goal of this work, interested readers are directed to [32,33].

### 1.3. Main Contributions

Previous research on the P2P community-based energy market focused only on distributed ledger technologies or on separating the market clearing problem from the OPF problem or else did not consider the network constraints at all.

In response to this shortcoming, this paper extends the current system of community energy trading by explicitly considering the underlying network constraints at the distribution level. To the best of our knowledge, this is the first model to merge network constraints in a community-based distributed P2P energy trading model. All transactions must be validated during the bidding process based on network constraints. In summary, the main contributions of this work are as follows:

- We designed a network-constrained community-based market in a single model with P2P trading and the OPF problem, to control voltage and capacity problems in the community network. The market framework is proposed and modeled in a centralized and distributed fashion. The market clearing on a day-ahead basis is performed accounting for actual energy exchanges, given the uncertainty in renewable generation.
- We demonstrate a specific distributed decomposition approach of the network-constrained community-based market using ADMM to solve the day-ahead market clearing problem, showing the feasibility of this model for P2P community-based energy trading schemes. The ADMM algorithm was decomposed based on consensus and exchange ADMM techniques and tested on a 10-bus medium voltage radial distribution network, to improve data privacy and reduce communication burden.

*1.4. Paper Structure*

The rest of this paper is structured as follows. In Section 2 the centralized network-constrained community-based market is introduced. Section 3 proposes the distributed approach of the network-constrained community-based market presented in Section 2. In Section 4 the case study is characterized and the numerical results for the distributed model are presented. Finally, the paper is concluded in Section 5.

## 2. Network-Constrained Community-Based Market

A community-based P2P market can be applied to collaborative micro-networks [34,35] and groups of neighboring peers [36], in which community fellows share common goods and plans. P2P energy exchange among community members is supervised by a community manager [37]. Note that peers need to share limited information with their supervisor to keep a high privacy level [28,38].

The integration of a community manager can facilitate the application of market regulation and the interface with the DSO. In the local market, the community manager identifies each prosumer as either a seller or a buyer, depending on the available generation and demand. Each seller offers his price per unit of electricity and the amount of electricity he wishes to sell from his preferred supplier to the market. Similarly, each buyer offers his preferred price and the amount of electricity he wishes to buy from the community. The community manager facilitates electricity trading by determining whether the sum of all sellers' electricity bids is sufficient to meet the demand of all buyers at the market clearing price, and advises participants to submit new bids if necessary. Once prosumers have completed intra-community trading, the community manager can accelerate bidding for the remaining excess electricity in the retail market. In a small community, the overall amount of surplus electricity fed into the grid is small. Therefore, it can be presumed that this will not affect the supply-demand balance and will not have a noteworthy impact on the market clearing price [39].

A detailed mathematical formulation of a network-constrained community-based market is presented in this section.

Consider a radial distribution network $G := (N^E, L^E)$, where $N^E$ denotes the nodes set and $L^E$ denotes the lines set. Besides the slack bus, every node has one parent node and a set of child nodes, represented by $C_i$. One can assume that every line points from a node $i$ to its parent node. Given a simple phase radial distribution network, the branch flow model developed in [40] was relaxed employing the Second Order Cone (SOC) convex relaxation [41].

The objective of the optimization problem is to minimize the total operating costs of the community, subject to the network's operational constraints, given by Equations (1a)–(1k).

$$\min_{\Gamma} \sum_t^T \left( \sum_i^{\Omega_i} \pi_{i,t} \cdot (P_{i,t}^{\mathrm{g}} - P_{i,t}^{\mathrm{l}}) - \sum_i^{\Omega_i} \pi_{i,t}^{\mathrm{SUR}} \cdot \beta_{i,t} + \sum_i^{\Omega_i} \pi_{i,t}^{\mathrm{SUP}} \cdot \alpha_{i,t} \right) \tag{1a}$$

$$s.t. \ \ P_{i,t}^{\mathrm{g}} - P_{i,t}^{\mathrm{l}} = q_{i,t}^{\mathrm{LEM}} + \beta_{i,t} - \alpha_{i,t}, \ \forall (i,t) \in (\Omega_i \backslash \{0\}, T) \tag{1b}$$

$$\sum_i^{\Omega_i} q_{i,t}^{\mathrm{LEM}} = 0 \ \ : \hat{\theta}_t, \ \ \ \forall t \in T \tag{1c}$$

$$P_{i,t}^{\mathrm{g}} - P_{i,t}^{\mathrm{l}} = \sum_{j \in C_i}^{N^E} P_{i,j,t}^{\mathrm{F}} - \sum_{j \in C_i}^{N^E} P_{j,i,t}^{\mathrm{F}} - R_{i,j} \cdot I_{j,i,t}, \ \forall (i,t) \in (N^E, T) \tag{1d}$$

$$Q_{i,t}^{\mathrm{g}} - Q_{i,t}^{\mathrm{l}} = \sum_{j \in C_i}^{N^E} Q_{i,j,t}^{\mathrm{F}} - \sum_{j \in C_i}^{N^E} Q_{j,i,t}^{\mathrm{F}} - X_{i,j} \cdot I_{j,i,t}, \ \forall (i,t) \in (N^E, T) \tag{1e}$$

$$V_{i,t} - V_{j,t} = 2 \cdot (R_{i,j} \cdot P_{i,j,t}^{\mathrm{F}} + X_{i,j} \cdot Q_{i,j,t}^{\mathrm{F}}) - I_{j,i,t} \cdot (R_{i,j}^2 + X_{i,j}^2), \ \forall (i,j,t) \in (N^E, N^E, T) \tag{1f}$$

$$V_{i,t} \cdot I_{i,j,t} \geq {P_{i,j,t}^{\mathrm{F}}}^2 + {Q_{i,j,t}^{\mathrm{F}}}^2, \ \forall (i,j), t \in L^E, T \tag{1g}$$

$$\alpha_{i,t}, \ \beta_{i,t}, \ \geq 0, \ q_{i,t} \in \mathbb{R}, \ \forall \ (i,t) \in (\Omega_i, T) \tag{1h}$$

$$\underline{V_{i,t}^{\min}} \leq V_{i,t} \leq \overline{V_{i,t}^{\max}}, \ \forall (i,t) \in (N^E, T) \tag{1i}$$

$$0 \leq I_{i,j,t} \leq \overline{I_{i,j,t}^{\max}}, \ \forall (i,j), t \in L^E, T \tag{1j}$$

$$0 \leq P_{i,t}^{\mathrm{g}}, Q_{i,t}^{\mathrm{g}} \leq \overline{P_{i,t}^{\mathrm{g}}}, \overline{Q_{i,t}^{\mathrm{g}}}, \ \forall (i,t) \in (N^E, T) \tag{1k}$$

where $\Gamma$ is the set of decision variables of the problem and for each peer $i$, the energy produced $P_{i,t}^{\mathrm{g}}$ and consumed $P_{i,t}^{\mathrm{l}}$ in the time interval $t$ is determined. We assume a 20% flexibility for each demand. Equation (1a) models the objective function of the problem. $\alpha_{i,t}$ is the imported energy priced according to the peer's $i$ retailer, $\pi_{i,t}^{\mathrm{SUP}}$. $q_{i,t}^{\mathrm{LEM}}$ are the P2P trades from the peer $i$ to the community. In addition, $\beta_{i,t}$ is the energy exported, which is priced in relation to the feed-in price in the period $t$, $\pi_{i,t}^{\mathrm{SUR}}$. The constraint (1b) concerns the energy balance for each peer $i$ for the respective time set $T$. The sum of internal transactions made in the community involving all community members must necessarily be zero, assured by constraint (1c).

Constraints (1d) and (1e) define the active and reactive power flow equations, respectively. Voltage drop along a distribution line is given by constraint (1f). Constraint (1g) defines the branch flow at the parent node of each line following the relaxed branch flow model. Constraints (1h) to (1k) establish the domain of the variables.

Note that for the application of this type of models, ICT and platform support is required to ensure communication between the different agents in the system. Then, we assume that platforms, as proposed in [42], can be used to support the proposed models.

## 3. Distributed Community-Based Market Approach

As mentioned in the literature review in Section 1.2, several approaches have solved community-based markets with OPF in a centralized fashion. However, the implementation of these methods requires the full information of all stakeholders, violating privacy terms. As mentioned earlier, the major contribution of this paper is the integration of an OPF problem with the P2P community-based market in a single distributed approach, resulting in a fully decentralized algorithm that maximizes social welfare. P2P exchanges and grid imports are minimized for all community members $i \in \Omega_i$ individually and in parallel accounting for network constraints.

The network-constrained community-based presented in Section 2 has several coupling constraints. For one, the community internal trades balance in constraint (1c) is a coupling constraint and has a dual variable $\hat{\theta}_{i,t}$ associated, which represents the community's internal trading price. Here, we face a sharing problem with an attractive economic interpretation known as the exchange problem [15]. The variable $\theta_{i,t}^{(k)}$ converges to an opti-

mal dual variable, easily interpreted as a set of optimal or clearing prices for the exchange. The proximal term in (2a) regarding the internal P2P trades is a penalty for $q_{i,t}^{\text{LEM}(k+1)}$ deviating from $q_{i,t}^{\text{LEM}(k)}$, projected onto the feasible set. This update in exchange ADMM can be carried out independently in parallel, for $i = 1, ..., \Omega_i$.

The goal of this algorithm is to solve the augmented Lagrangian relaxation of the coupling constraints by an iterative and collaborative process to find the optimal solution to the original problem while decomposing the problem. If the coupling constraint is an equilibrium equation, one obtains a special case of the canonical problem known as the sharing problem. In this paper, the general consensus form of the ADMM algorithm is applied to solve the community-based market considering network constraints in a distributed manner.

Figure 1 presents the proposed structure for the distributed community-based market model where each community agent communicates with their adjacent neighbors and the associated ADMM coordinator. In this type of structure, each agent is responsible for their energy exchanges, either within the community or outside it. Each agent is solely responsible for updating the ADMM coordinator of the exchanges it intends to carry out in the market after solving its cost optimization problem. In this way, all agents are connected to the ADMM coordinator and at the same time to the community network. In this structure, there is no community manager where the agent has to communicate the exchanges he intends to make outside the community. In this way, this structure is more simplified and with less sharing of information by his participants.

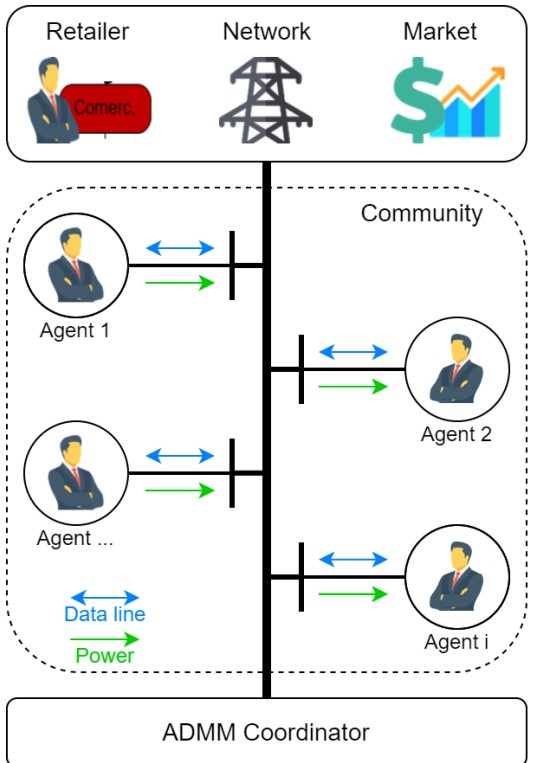

**Figure 1.** Decentralized community-based market structure.

The centralized problem in (1) is reformulated creating an individual subproblem for every node/agent *i*. The solutions must agree at the global optimum through replicated local coupling variables. At each ADMM iteration, the ADMM coordinator checks for convergence through the balance of the coupling variables and updates the dual variables for the following iteration with the penalty parameter $\rho$. Consensus is reached when the relevant local copy for each subproblem matches with the global coupling variable. The local copy of the coupling variables passed for each subproblem is considered as a

parameter of the optimization problem solved by node $i$ at the current iteration $k$, denoted by a hat. In (2a) is the market clearing subproblem for each node, and in (2b) are presented the constraints of the decomposed model which are some of the constraints presented in the model (1).

$$
\begin{aligned}
\min_{\Gamma} \quad & \sum_{t}^{T} \Bigg( \pi_{i,t} \cdot \left( P_{i,t}^{g^{(k)}} - P_{i,t}^{l^{(k)}} \right) - \pi_{i,t}^{\text{SUR}} \cdot \left( \beta_{i,t}^{(k)} + \sum_{j \neq i}^{\Omega_i} \hat{\beta}_j^{(k-1)} \right) + \pi_{i,t}^{\text{SUP}} \cdot \left( \alpha_{i,t}^{(k)} + \sum_{j \neq i}^{\Omega_i} \hat{\alpha}_j^{(k-1)} \right) + \\
& \hat{\theta}_{i,t}^{(k-1)} \cdot q_{i,t}^{\text{LEM}^{(k)}} + \frac{\rho}{2} \left\| q_{i,t}^{\text{LEM}^{(k)}} - \left( \hat{q}_{i,t}^{\text{LEM}^{(k-1)}} - \frac{1}{\Omega_i} \cdot \sum_{i,t}^{\Omega_i} \hat{q}_{i,t}^{\text{LEM}^{(k-1)}} \right) \right\|^2 + \\
& \sum_{j \in C_i} \left( \hat{\lambda}_{j,t}^{(k-1)} \cdot \left( P_{j,t}^{F^{(k)}} - \hat{P}_{j,t}^{F^{(k-1)}} \right) + \frac{\rho}{2} \left\| P_{j,t}^{F^{(k)}} - \hat{P}_{j,t}^{F^{(k-1)}} \right\|^2 \right) + \qquad\qquad , \forall i \in \Omega_i \\
& \sum_{j \in C_i} \left( \hat{\delta}_{j,t}^{(k-1)} \cdot \left( Q_{j,t}^{F^{(k)}} - \hat{Q}_{j,t}^{F^{(k-1)}} \right) + \frac{\rho}{2} \left\| Q_{j,t}^{F^{(k)}} - \hat{Q}_{j,t}^{F^{(k-1)}} \right\|^2 \right) + \\
& \sum_{j \in C_i} \left( \hat{\zeta}_{j,t}^{(k-1)} \cdot \left( I_{j,t}^{(k)} - \hat{I}_{j,t}^{(k-1)} \right) + \frac{\rho}{2} \left\| I_{j,t}^{(k)} - \hat{I}_{j,t}^{(k-1)} \right\|^2 \right) \Bigg)
\end{aligned}
\tag{2a}
$$

$$
\text{s.t.} \quad \text{(1b), (1d), (1e), (1h)–(1k)}
$$

$$
V_{i,t}^{(k)} - \hat{V}_{j,t}^{(k-1)} = 2 \cdot \left( R_{i,j} \cdot P_{i,j,t}^{F^{(k)}} + X_{i,j} \cdot Q_{i,j,t}^{F^{(k)}} \right), \ \forall (j,t) \in (N^E, T)
\tag{2b}
$$

where $\rho$ represents the penalty parameter which is predefined and $\theta, \lambda, \delta, \zeta$ are the dual variables. The ADMM algorithm will iterate through the steps and the dual variables updated at each step. The updates of the dual variables are shown in Equations (2c)–(2f). In the case of convex problems, dual variables will converge to the dual values associated with the constraints of the centralized approach [15]. In that case, the problem is continuous and convex. This technique allows for clearing the proposed network-constrained community-based market in a distributed fashion with optimality guarantees.

$$
\theta_{i,t}^{(k)} = \theta_{i,t}^{(k-1)} + \rho * \left( \frac{1}{\Omega_{i,t}} \cdot \sum_{i,t}^{\Omega_{i,t}} \hat{q}_{i,t}^{\text{LEM}^{(k)}} \right)
\tag{2c}
$$

$$
\lambda_{i,j,t}^{(k)} = \lambda_{i,j,t}^{(k-1)} + \rho * \left( P_{i,j,t}^{F^{(k)}} - \hat{P}_{i,j,t}^{F^{(k-1)}} \right)
\tag{2d}
$$

$$
\delta_{i,j,t}^{(k)} = \delta_{i,j,t}^{(k-1)} + \rho * \left( Q_{i,j,t}^{F^{(k)}} - \hat{Q}_{i,j,t}^{F^{(k-1)}} \right)
\tag{2e}
$$

$$
\zeta_{i,j,t}^{(k)} = \zeta_{i,j,t}^{(k-1)} + \rho * \left( I_{i,j,t}^{(k)} - \hat{I}_{i,j,t}^{(k-1)} \right)
\tag{2f}
$$

The ADMM algorithm will iterate through the step until convergence is reached. These conditions are evaluated and defined by the Equations (2g)–(2k).

$$
\begin{aligned}
& \sum_{t}^{T} \sum_{i}^{\Omega_i} \left( \pi_{i,t}^{\text{SUP}} \cdot q_{i,t}^{\text{LEM}^{(k)}} - \pi_{i,t}^{\text{SUR}} \cdot \beta_{i,t}^{(k)} + \pi_{i,t}^{\text{SUP}} \cdot \alpha_{i,t}^{(k)} \right) - \\
& \left( \pi_{i,t}^{\text{SUP}} \cdot q_{i,t}^{\text{LEM}^{(k-1)}} - \pi_{i,t}^{\text{SUR}} \cdot \beta_{i,t}^{(k-1)} + \pi_{i,t}^{\text{SUP}} \cdot \alpha_{i,t}^{(k-1)} \right) \leq \epsilon, \ \forall i \in N^E
\end{aligned}
\tag{2g}
$$

$$
\frac{\sum_{t}^{T} \sum_{i}^{\Omega_i} q_{i,t}^{\text{LEM}^{(k)}}}{\Omega_i} \leq \epsilon^{\theta}
\tag{2h}
$$

$$
P_{i,j,t}^{F^{(k)}} - P_{j,i,t}^{F^{(k)}} \leq \epsilon
\tag{2i}
$$

$$
Q_{i,j,t}^{F^{(k)}} - Q_{j,i,t}^{F^{(k)}} \leq \epsilon
\tag{2j}
$$

$$
I_{i,j,t}^{F^{(k)}} - I_{j,i,t}^{F^{(k)}} \leq \epsilon
\tag{2k}
$$

where $\epsilon$ is the authorized tolerance for the primal and dual residuals, usually set with a low value such as $10^{-4}$. When these criteria become less or equal than the specified tolerance the ADMM algorithm stops and has converged. Besides meeting the convergence conditions the ADMM algorithm is also limited to a certain number of iterations.

Figure 2 illustrates how the network-constrained community-based market centralized market is decomposed in $\Omega_i$ subproblems, where each is independently by each community member. The iterative process continues until the convergence criteria is reached. A common convergence criterion is that the primary residual is less than a certain threshold $\epsilon$, as expressed in (2g). Furthermore, the dual convergences are verified for each dual variable as in (2h)–(2k). If convergence is met the iterative process ends, if not, the iterative process continues and the coupling variables are updated through the ADMM agent which updates every agent who is responsible to run his individual market clearing problem.

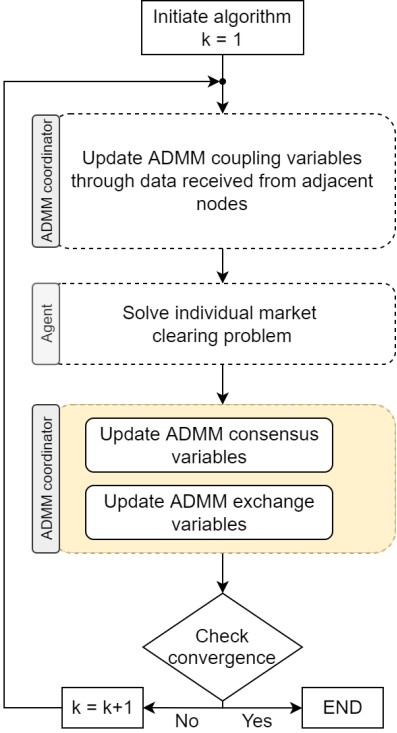

**Figure 2.** ADMM framework for the network-constrained community-based market.

## 4. Case Study

The case study aims to test and validate the proposed mathematical model under different economic approaches for sharing assets in an energy community.

### 4.1. Case Characterization

The case is based on a single community with two different types of participants, consumers, and prosumers. Data from five different patterns for housing and commercial buildings is used to run the simulation over one day. The retail price and the grid access tariff price for the allocated energy are established according to Portugal's time-of-use (ToU) tariff, following the Energy Services Regulatory Entity [43]. The community comprises 9 agents in a 10-bus radial distribution system from MatPower [44], illustrated in Figure 3. Node 0 is used as the reference bus and represents a link to the external supplier, where community agents situated in nodes 1 to 9 can import or export energy. Each node from 1 to 9 has a deterministic load profile. Solar arrays were randomly placed at 40% of the nodes. The nodes with PV systems are nodes 1, 4, 7, and 9. Note that the squared voltage $V_{i,t}$, varies between 5% of the nominal value defined as 1 p.u. for all buses, and the slack bus is fixed at 1 p.u. for the entire time set.

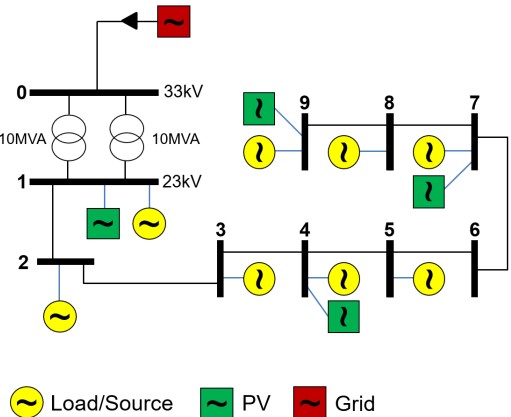

**Figure 3.** Electric diagram of the 10-bus MV radial distribution system.

The input data related to the agent's power production/consumption and detailed tariff prices are available at Mendeley Data [45].

*4.2. Results*

In this section, the proposed network-constrained community-based market is tested on the 10-bus test system. The modelling was conducted in Python with an open-source optimization modeling language, Pyomo [46], with IPOPT [47] as a MILQP solver on an AMD Ryzen 5 PRO 4650U 2.10 GHz processor with 16GB RAM. The simulations were made for the first day of 2021 ($T = 24$).

The ADMM algorithm converged in 1632 iterations using a convergence penalty term $\rho = 1.0$ in a total of 1670.2 s. The convergence is reached when the P2P internal trades error is less or equal to $\epsilon^\theta = 0.006$ and the primal residual along with the other residuals less or equal to $\epsilon = 0.0001$.

The effects of the network topology can be seen in the voltage of the individual nodes (see Figure 4). In the absence of current flow constraints, the upper voltage limit at the slack bus (node 0) becomes the binding constraint in the optimal case, set at 1.00 pu; voltages at all other nodes drop with distance from the feeder due to line effects, or increase when prosumers supply the community or export to the grid. General voltage trends can be seen throughout the day, with a considerable drop in hours 18 through 23 when the sunset and peak inflexible load result in a spike in net load across the grid. The effects of DER scheduling are also visible on some nodes, particularly nodes 8 and 9, where community representatives act as generators and intend to trade energy between other community members or export energy to the grid.

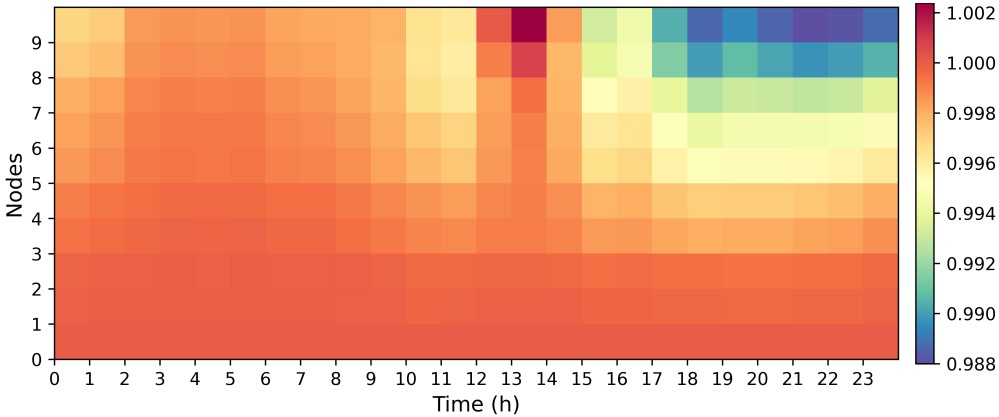

**Figure 4.** Nodes voltage results for the centralized model.

A comparison was made between the results of nodes' voltages of the centralized model, described in Section 2, and the decentralized model, described in Section 3, and the biggest difference observed between the magnitudes was 0.0009864, that is, an error of 0.098%.

In Table 2 are summarized the results for the simulated centralized and distributed models. The total amount of energy internally traded in the community, the social welfare (SW), and some of the agent's costs in the community are shown for the sake of space. As one can see, the results gathered from the distributed model are very close to the ones from the centralized model. However, a bigger gap can be seen for the P2P trades, where the distributed model faces more difficulties to reach convergence.

**Table 2.** Community P2P trades, social welfare, and agents' costs/revenues for both centralized and distributed models.

| Model | P2P Trades | SW | Agents' Costs/Revenues | | | | |
|---|---|---|---|---|---|---|---|
| | | | 1 | 2 | 3 | ... | 9 |
| Centralized | 0.6902 | 1.6455 | 0.131 | 0.231 | 0.301 | ... | 0.148 |
| Distributed | 0.4839 | 1.6507 | 0.145 | 0.234 | 0.299 | ... | 0.139 |

In addition, the convergence results of the ADMM algorithm, the primal and dual residuals, are shown in Figures 5 and 6. The obtained results show that the presented procedure reaches convergence with a 0.0006 error for the P2P internal trades and 0.0001 for the primal residual and active and reactive power trades residuals after 1600 iterations. Based on the convergence metrics, the algorithm has converged with reasonable optimality in almost 1600 iterations. As one can see through Figures 5 and 6 the blue line representing the P2P internal trades balance of the community, presented in Equation (1c), is the residual that has the small convergence rate limiting the algorithm to reach convergence sooner at the 950th iteration to the 1632nd iteration. One can also see that the primal residual, in Equation (2g), has the fastest convergence rate followed by the $\delta$ residuals and $\lambda$ residuals, which are associated with the reactive power trades and active power trades, presented in Equations (2j) and (2i), respectively.

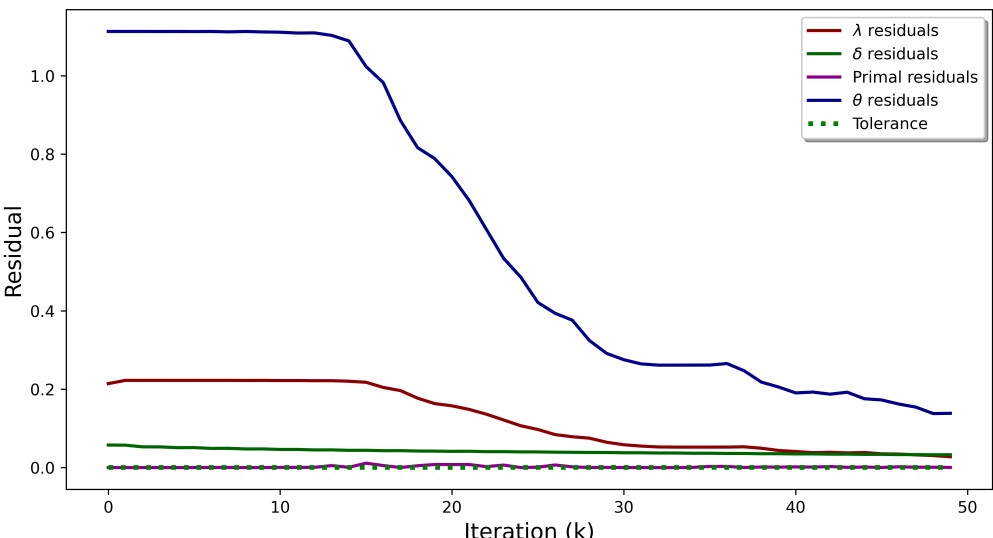

**Figure 5.** ADMM dual variables convergence rates from iteration 0 to 50.

The LEM price with the community tariffs, resulting from the market clearing problem, is shown in Figure 7. The algorithm is conducted in a fairly small and homogeneous setting. Hence, the prices in the LEM behave as predicted. When there are more PV generation in the

community the internal price is lower. One can see that during the daytime, when trading should arise, the LEM price is lower than the market price, but higher than the feed-in tariff.

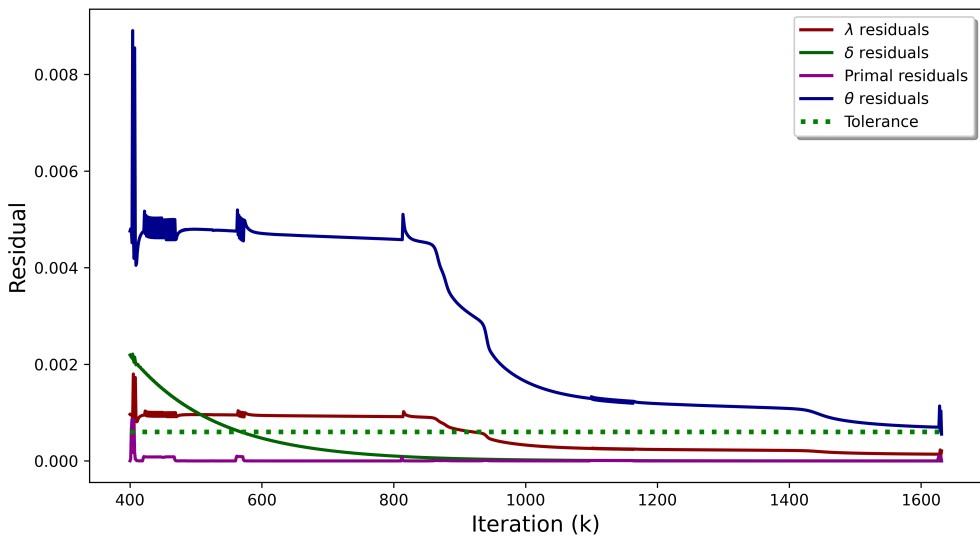

**Figure 6.** ADMM dual variables convergence rates from iteration 400 to 1632.

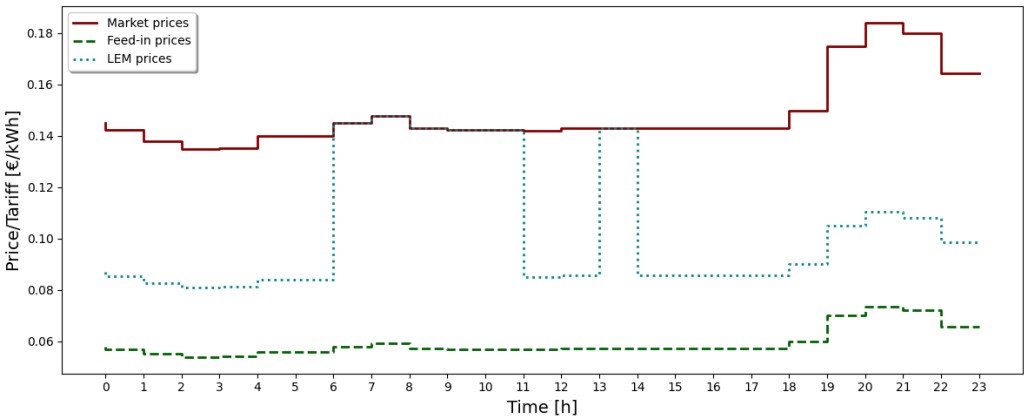

**Figure 7.** Community tariffs and prices for the simulated period.

In Figure 8 imports, exports, and P2P internal trades of the community for the simulated period are presented in stacked bars. As one can see, the demand is always met, and the P2P internal exchanges are made in favorable periods for PV. The periods in which more internal exchanges occur in the community correspond to periods where the demand is not very high and in periods that are quite favorable for photovoltaic production.

As one can see, most of the community's demand is met through imports. A total of 95.5% of the community's demand comes from imports from the external supplier, and the remaining 4.5% from internal exchanges in the community through surplus PV from prosumers. No exports are made by the community for the modeled case study. One can see that from hours 7 to 14 there are P2P exchanges. These internal exchanges in the community are made between a prosumer (who at that moment can be considered a producer) and a consumer. Considering that PV systems reach their production peaks between 10–14 h, it is natural that there may be internal exchanges in this period. It is when there is an increase in internal exchanges in the community. It should be noted that the volume of exchanges also depends on the demand of each member of the community. If the demand is greater than his PV generation, this member will not be a prosumer but a consumer. Thus, peaks of P2P exchanges occur at hours 12, 13, and 14, which is when the volume of PV generation is high and the energy demand is relatively low.

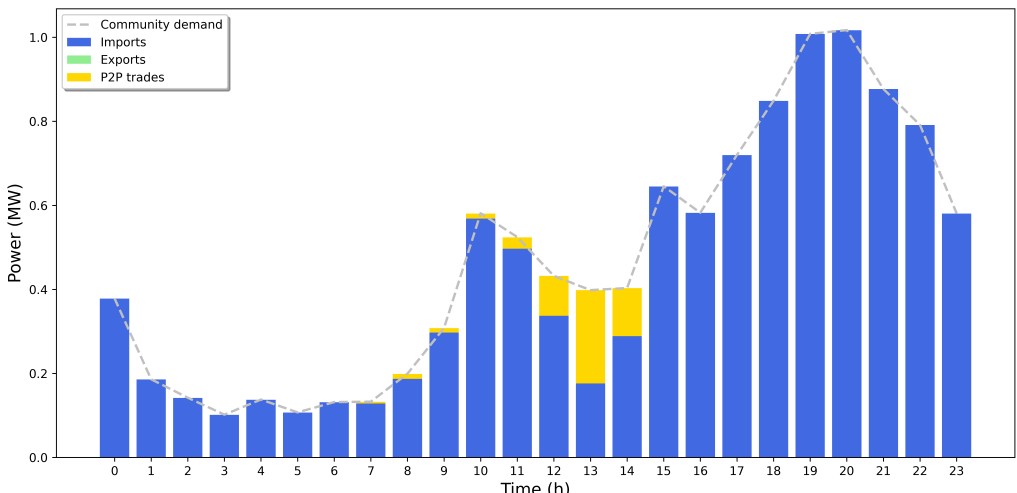

**Figure 8.** P2P internal trades, imports, and exports for the simulated period.

### 4.3. ADMM Scenarios

For the first test case, we created five scenarios based on the penalty factor $\rho$. The algorithm is tested for each one of them. The five scenarios depend on the number of community agents of the instance. Therefore the scenarios are: $\rho = 1/\Omega_i$, $\rho = 1/\sqrt{\Omega_i}$, $\rho = 1$, $\rho = \Omega_i/4$ and $\rho = \Omega_i/2$. We run the algorithm until the residual conditions are verified following Equations (2g)–(2k), or the limit of 2000 iterations is reached. The outcomes of the model are shown in Table 3.

**Table 3.** Penalty term ($\rho$) optimal scenario results.

| $\rho$ | Iterations | Objective Value | Primal Residual | Dual Residuals | | |
|---|---|---|---|---|---|---|
| | | | | $\lambda$ | $\delta$ | $\theta$ |
| $1/\Omega_i$ | 2000 | 0.257694 | $\approx 0$ | 0.000576 | 0.000001 | 0.002591 |
| $1/\sqrt{\Omega_i}$ | 2000 | 0.258385 | 0.001325 | 0.001288 | 0.000013 | 0.000531 |
| 1 | 1632 | 0.260329 | 0.000091 | 0.000023 | 0.000001 | 0.000554 |
| $\Omega_i/4$ | 2000 | 0.264446 | 0.000001 | 0.000451 | 0.000001 | 0.002031 |
| $\Omega_i/2$ | 2000 | 0.263014 | 0.000136 | 0.000330 | 0.000001 | 0.000747 |

The scenario where $\rho = 1$ reveals the best performance, providing the quickest convergence. It needed 1632 iterations to converge while the other scenarios did not converge reaching the limit of 2000 iterations stipulated for the algorithm. However, comparing the objective values reached at the end of the algorithm one can see that there is a small gap for the values shown. As already mentioned the $\delta$ residuals are the ones reaching and as one can see, the dual variable $\delta$ is the only one that converges for all scenarios. Since the other scenarios, besides $\rho = 1$, did not converge we cannot really make an association for the residuals shown in the table because those are the ones presented for the final iteration.

In addition to the scenarios with the penalty term, we also simulated the distributed model without OPF, that is, without network constraints. The comparison of models with and without OPF is presented in Table 4.

As one can see, the distributed model without network constraints converges considerably faster and requires 200 iterations to converge to the optimal solution of the problem. This scenario presents a lower social welfare, which means in this case that the total costs of the community have gone down. This is justified by the fact that more energy is exchanged internally in the community, about 5%, equivalent to 25.4 kW.

**Table 4.** ADMM scenarios with and without OPF.

|  | Social Welfare | P2P Trades | Iterations | Computational Time |
|---|---|---|---|---|
| With OPF | 1.6507 | 0.4839 | 1632 | 1670.2 s |
| Without OPF | 1.6304 | 0.5093 | 219 | 200.7 s |

The proposed ADMM framework for this case study has the pros:

- to divide and conquer, splitting our large problem into a series of subproblems, where each subproblem is solved individually per community agent;
- Data distribution. The splitting method of the master problem can be applied to large-scale data, distributing the data across different local problems and optimizing objectives locally with some communication on the dual variables between the subproblems;
- Robust. It has a very little assumption on the property of the objective function, converges with good precision; Easy to implement. Reasonably easy to reach to needed ADMM algorithm for the market problem.

The main disadvantage of the ADMM framework is related to its convergence rate. For this specific case study, the ADMM algorithm took 1632 iterations to converge to high accuracy. According to Stephen et al. [15], simple examples show that ADMM can converge very slowly to reach high-accuracy solutions. However, to reach modest accuracy, ADMM can converge within a few tens of iterations. Its slow convergence rate differentiates it from other algorithms such as Newton's method where high accuracy can be reached in a fair amount of time.

## 5. Conclusions

Modeling and simulation of P2P markets is vital for studying and planning there integration in the distribution network, especially for regulatory standards and proper market designs. The goal of this paper was to review P2P markets integrated in a distribution network works and model a distributed P2P community-based market with network constraints in a single optimization problem. The review found that the network constraints are not considered in most studies concerning P2P community markets.

A network-constrained P2P community-based market structure is presented and modeled in a centralized and distributed fashion. Market clearing on a day-ahead basis is performed considering P2P trading, as renewable generation is uncertain. We explicitly considered the effects of injecting and absorbing energy of the community internal trades in the network. A distributed market clearing algorithm is presented to improve privacy and reduce communication overhead. The algorithm shows promising convergence and scaling properties for practical implementation in the case under study. This algorithm was decomposed based on consensus and exchange ADMM techniques and tested on a 10-bus medium voltage radial distribution network. The distributed process reduces the quantity of information shared. The information that each community member must share is the profile of P2P exchanges with the ADMM coordinator and with the other community neighbors for updating the multipliers at each iteration and for convergence assessment of the coupling variables.

The results obtained with the distributed approach were compared with those obtained with the central approach. Both the centralized and distributed approaches yield comparable results with an acceptable computational cost. The values of the objective function, the social welfare, the agents costs and the profiles of the power flow at the connection with the external grid (external supplier) match. The results show that the proposed market reduces users' energy costs and establishes a local balance between household generation and demand without violating the network constraints. The comparison from the centralized and distributed ADMM approach shows an 0.098% error for the nodes' voltages, where the best ADMM convergence rate was achieved with the penalty term at 1 among five different scenarios. The integrated OPF in the community-based market is a computational burden that increases the resolution of the market dispatch problem by

about 8 times the computation time, from 200.7 s (without OPF) to 1670.2 s. The P2P internal trades balance of the community is the residual that has a small convergence rate limiting the algorithm to reach convergence sooner at the 950th iteration to the 1632nd iteration.

Future work could investigate the sensitivity of the modeling results to input parameters such as trade coefficients, investment costs, and DER distribution. In addition, a detailed techno-economic evaluation could be conducted to assess social welfare and convergence over a longer time period.

**Author Contributions:** Conceptualization, C.O., M.S, L.B., T.S. and M.A.M.; Data curation, C.O. and M.S.; Formal analysis, C.O., M.S., L.B. and T.S.; Investigation, C.O., M.S. and L.B.; Methodology, C.O., M.S. and L.B.; Supervision, T.S. and M.A.M.; Validation, C.O., M.S. and L.B.; Writing—original draft, C.O. and M.S.; Writing—review & editing, C.O., M.S., L.B., T.S. and M.A.M.; All authors have read and agreed to the published version of the manuscript.

**Funding:** This research received partial support from the Federal University of Juiz de Fora (UFJF), National Council for Scientific and Technological Development (CNPq), Coordenação de Aperfeiçoamento de Pessoal de Nível Superior—Brasil (CAPES)—Finance Code 001, Fundação de Amparo à Pesquisa no Estado de Minas Gerais (FAPEMIG) and INERGE. It is also supported by Norte Portugal Regional Operational Programme (NORTE 2020), under the PORTUGAL 2020 Partnership Agreement, through the European Regional Development Fund (ERDF), within the DECARBONIZE project under agreement NORTE-01-0145-FEDER-000065 and by the Scientific Employment Stimulus Programme from the Fundação para a Ciência e a Tecnologia (FCT) under the agreement 2021.01353.CEECIND.

**Data Availability Statement:** Not applicable.

**Conflicts of Interest:** The authors declare no conflict of interest.

## Nomenclature

### Parameters

| | |
|---|---|
| $P_{i,t}$, $\overline{P_{i,t}}$ | Lower and upper bounds for peer $i$ in period $t$; |
| $\pi_{i,t}^{\mathrm{SUP}}$ | Price of the energy supply from peer's $i$ retailer in period $t$; |
| $\pi_{i,t}^{\mathrm{SUR}}$ | Price of the feed-in energy of peer $i$ in period $t$; |
| $P_{t,i}^{\mathrm{l}}$, $Q_{t,i}^{\mathrm{l}}$ | Active/Reactive power demand at bus $i$ in period $t$; |
| $\pi_{t,i}^{\mathrm{g}}$ | Generation cost at bus $i$ in period $t$; |
| $R_i$, $X_i$ | Resistance/Reactance of line $i$ in period $t$; |
| $\overline{P_{i,t}^{\mathrm{g}}}$, $\overline{Q_{i,t}^{\mathrm{g}}}$ | Active/Reactive power generation limit. |

### Variables

| | |
|---|---|
| $P_{i,t}$ | Power produced/consumed by peer $i$ in period $t$; |
| $\alpha_{i,t}$, $\beta_{i,t}$ | Electricity imported/exported from the main grid by peer $i$ in period $t$ through the energy community; |
| $q_{i,t}$ | Internal trade in the community by peer $i$ in period $t$; |
| $P_{t,i}^{\mathrm{F}}$, $Q_{t,i}^{\mathrm{F}}$ | Active/Reactive power injected in line $i$ in period $t$; |
| $P_{t,i}^{\mathrm{g}}$, $Q_{t,i}^{\mathrm{g}}$ | Active/Reactive power generation at bus $i$ in period $t$; |
| $V_{t,i}$ | Squared voltage magnitude at bus $i$ in period $t$; |
| $I_{t,i}$ | Squared current magnitude at line $i$ in period $t$. |

### Sets

| | |
|---|---|
| $\Omega_i$ | Set of peers $n$; |
| $T$ | Set of time periods; |
| $N^E$ | Set of buses/nodes of electricity network; |
| $L^E$ | Set of lines of of electricity network. |

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
