# Peer review of "Distributed Network-Constrained P2P Community-Based Market for Distribution Networks"

_energies, doi:10.3390/en16031520_

Round 1

Reviewer 1 Report

Dear authors,

Please deal with the following concerns and address the requested items with appropriate explanations and justifications to clarify the ambiguities and amend mistakes. 

- Decentralized community-based market structure must be further explained.

- What are the pros and cons of the ADMM framework for the network-constrained community-based market? 

- What are the benefits and drawbacks of bilateral Energy trading, a Fully P2P market, a Microgrid scheme, and a community-based market to provide a comparison? Please add more explanation. 

- The interfaces required to connect multiple peers to each other in a fully P2P, community P2P, and hybrid P2P models need to be further explained.

- With regard to a high level of micro-transactions, do you believe that the current banking system can support the scheme properly, or is there a need for cryptocurrency?

- The interactions between community-scale P2P models with the distribution operator and retail market should be further explained. 

- What are the main reasons for the large contribution of the P2P scheme in demand satisfaction with hours 12, 13, and 14?

Author Response

Dear Reviewer,

Thank you for the comments and suggestions to improve our work.

All the comments have been considered. We kindly redirect the reviewer to the enclosed file that contains the details of all answers,

Thank you in advance,

The Authors

Reviewer 2 Report

This paper investigated the optimization of P2P energy market, which is an interesting topic and practically useful.

1) It would be better if some typical numerical results can be added in abstract or conclusion sections, which could highlight the findings or contributions of this research.

2)The robust optimization [R1] and stochastic optimization [R2] algorithms have been widely used for power system applications, which are recommended to add in background or literature review section.

[R1] Fang X, Du E, Zheng K, et al. Locational pricing of uncertainty based on robust optimization. CSEE Journal of Power and Energy Systems, 2020, 7(6): 1345-1356.

[R2] Xiao D, Qiao W. Hybrid scenario generation method for stochastic virtual bidding in electricity market[J]. CSEE Journal of Power and Energy Systems, 2021, 7(6): 1312-1321.

3)The incentives of using ADMM in this paper can be further introduced

4)The main contributions of this work are recommended to further explained in Section 1.3 on Page 4 

Author Response

(The authors gave the same response as above.)

Author Response

(The authors gave the same response as above.)
